# Experimental study of land subsidence in response to groundwater withdrawal and recharge in Changping District of Beijing

Yanbo Cao[1], Ya-ni Wei[1]*, Wen Fan[1,2], Min Peng[2], Liangliang Bao[3]

**1** School of Geology Engineering and Geomatics, Chang'an University, Xi'an, Shaanxi, China, **2** China Electronic Research Institute of Engineering Investigations and Design, Xi'an, Shaanxi, China, **3** School of Civil Engineering, Yulin University, Yulin, Shaanxi, China

* weiyani2006@126.com

**Data Availability Statement:** All relevant data are within the manuscript.

**Funding:** This study was supported by the Department of Science and Technology of Shaanxi

## Abstract

Over exploitation of groundwater in Changping District of Beijing city has caused serious land subsidence in the past decades. In recent years, the operation of the South-to-North Water Transfer Project has reduced the land subsidence rate. In this paper, Experimental tests are performed using the GDS Consolidation Testing System to characterize the compression and rebound of soils at depths of less than 100 m caused by groundwater withdrawal and recharge in Changping District. The results indicate that the compressible layers are the main contributors to land subsidence. The first compressible layer experiences greater deformation and more considerable hysteresis than the other compressible layers with the same decrease in the pore water pressure. Therefore, the exploitation of the adjacent aquifer should be controlled in the future. The deformation in the second and third compressible layers is a gradual and long-term process with little rebound; therefore, the subsidence should be seriously addressed when the groundwater in the two compressible layers is exploited on a large scale. In the same compressible layer, silty clay is more compressible and hysteretic than silt. For the same soil sample, the deformation rate decreases gradually as the pore water pressure decreases, whereas the creep deformation shows an overall increasing trend. A parameter named the subsidence index $C_w$ is proposed in this paper to describe the soil compressibility during groundwater withdrawal. All the soil samples are characterized by elastic-plastic deformation, and the shallow soil samples with less pore water pressure decrease are more likely to rebound.

## 1. Introduction

Land subsidence induced by groundwater withdrawal has been a worldwide problem [1–3], and more than 60 countries around the world are facing issues associated with this problem [4]. Land subsidence usually leads to damage to the aquifer system, decrease in water quality, and destruction of subsurface and surface structures, such as underground tunnels, buildings, roads, railways and pipelines [5–9]. There are three main subsidence-affected regions in

Province (Grant No. 2019SF-233, and 2017JM5144) to Min Peng and Liangliang Bao. This study was supported by Science and Technology Bureau of Yulin city (Grant No. 2014cxy-04) to Yanbo Cao. The funders had no role in study design, data collection and analysis, decision to publish, or preparation of the manuscript.

**Competing interests:** The authors have declared that no competing interests exist.

China, including the Yangtze River Deltaic Plain, North China Plain and Fenhe-Weihe Basin, which include more than 90 cities, such as Shanghai, Beijing and Xi'an [10–14].

Research on land subsidence has mainly focused on two aspects: land subsidence monitoring and land subsidence prediction. Land subsidence monitoring uses various techniques, such as interferometric synthetic aperture radar (InSAR) [15–17], leveling surveys, borehole extensometers [18,19] and GPS data [20,21] to characterize the spatial and temporal distribution of subsidence [22–28]. Land subsidence prediction uses statistical modeling, numerical simulation and artificial intelligence. Numerical models range from simple 2D seepage models to 3D fully coupled models that consider both 3D seepage and 3D consolidation [14, 29–32]. These models require long-term observation data for calibration and validation and estimation of the boundary conditions and parameters, such as the permeability and storage coefficient. Artificial intelligence (AI) techniques, including the artificial neural network (ANN) method and machine learning algorithms (MALs), are an effective approach to overcome the above challenge and have been widely used to predict land subsidence in recent years [33–35].

However, few studies have used experimental methods to characterize and predict land subsidence. Li et al. [36] performed field pumping well tests in the Pudong New Area of Shanghai to investigate the response of groundwater level and layer deformation. The results show that the soil deformation matches well with the groundwater fluctuation, and there is a close hydraulic connection between the pumping aquifer and the underlying aquifer. The largest subsidence occurs at the top of the pumping aquifer instead of the ground surface. Li et al. [37] conducted a physical model test on remolded sands with different initial densities to study soil deformation under cyclic withdrawal and recharge conditions and found that the deformation behaviors depend on both the initial density and number of withdrawal-recharge cycles. The sand initially exhibits elastic deformation and then exhibits plastic deformation with further withdrawal-recharge cycles. Such in situ or physical model experimental studies on land subsidence are usually expensive and take a long time.

In this paper, a laboratory-scale model test is designed using the GDS Consolidation Testing System to study land subsidence by simulating groundwater withdrawal and recharge. This system can reflect the deformation behaviors of undisturbed soils under actual stress conditions and is less costly in terms of financial resources and time than other methods. As the political and cultural center of China, Beijing has suffered from land subsidence since the 1950s along with the increasing urban expansion and great demands for water resources, and the subsidence has showed a rapid increase in the past several decades. Changping District, located in the northwest, is one of the emergency groundwater resource regions for Beijing. The long-term overexploitation of groundwater has made this district a major region for land subsidence, and the area of Shahe-Baxianzhuang in the district is one of the five major depression areas in Beijing. Until 2005, the depression area with cumulative settlement greater than 350 mm amounted to 97 km$^2$, and the settlement continued to increase linearly in the period of 2005 to 2011 [38]. In recent years, the South-to-North Water Transfer Project has obviously changed the structure of the water supply in Beijing [39]. The plan of limiting groundwater exploitation was advanced by relevant authorities, and most pumping wells in the emergency groundwater resource regions were sealed to control land subsidence [40]. As a result, the land subsidence rate has presented an overall decreasing trend. However, the rebound deformation of compressible layers that have been consolidated due to withdrawal is unclear. The layer response to recharge varies with the soil lithology, depth and degree of compaction. Therefore, this paper focuses on Changping District and aims to characterize and predict the soil compaction and rebound behaviors in different compressible layers under groundwater withdrawal and recharge conditions.

## 2. Geological and hydrogeological settings of the study area

In Changping District, the terrain is high in the northwest and low in the southeast. Mountains and plains are the two primary geomorphic types, with elevations varying from 800 to 1000 m and from 30 to 100 m, respectively. The surface sediments of the entire district consist mostly of Quaternary deposits with thicknesses ranging from 100 m to 600 m (Fig 1), which act as a natural reservoir of groundwater. The water-bearing formation evolves from a single layer composed of gravels in the northwest to an alternating layered system composed of sands, silts and clays in the southeast. Groundwater is the primary water resource for daily life and industry in the district. Precipitation is the main recharge for groundwater, and artificial exploitation is the primary discharge way. Influenced by the topography in the district, groundwater generally migrates from northwest to southeast, and the direction changes in local areas due to concentrated exploitation. Referring to the monitoring data of Water Bureau of Changping District [41], the shallow groundwater depth in the monitoring point of Machikou (marked in Fig 1) presented an increasing trend from 6.09 m in 2001 to 28.09 m in 2011. During the 10 years, the decline of groundwater level is more than 20 m, and the decline rate is an average of 2 m per year.

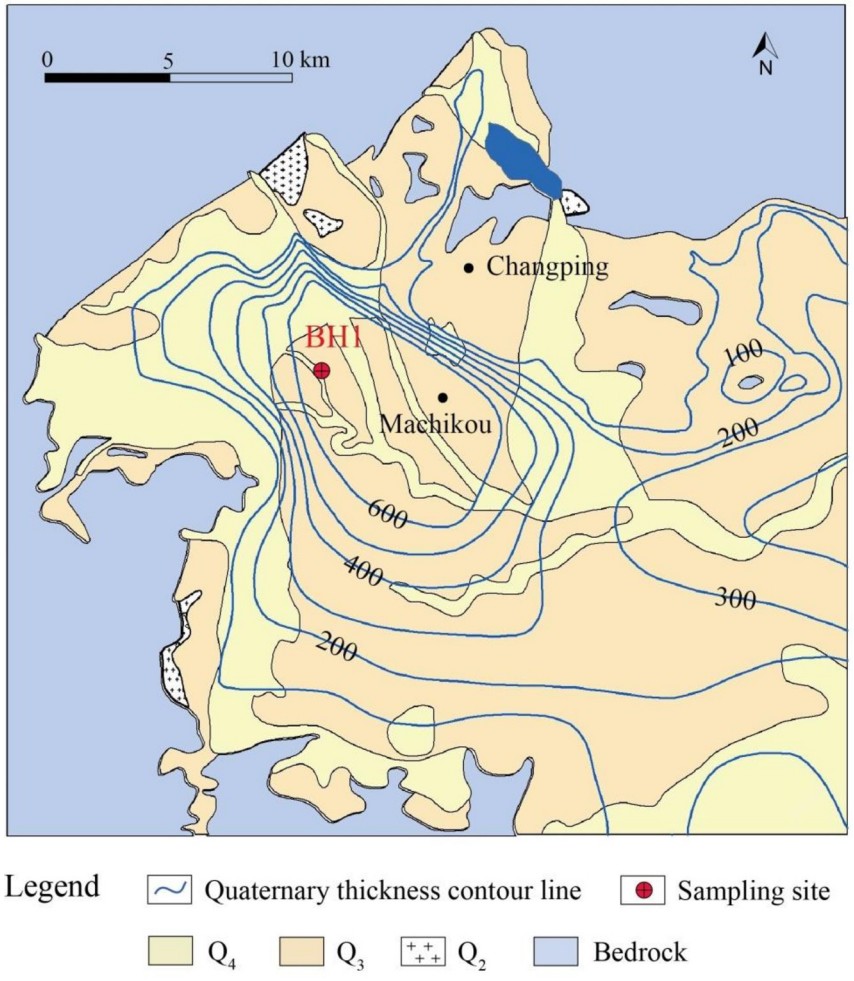

**Fig 1. Geological map of the study area (sampling site is located in the borehole BH1).**

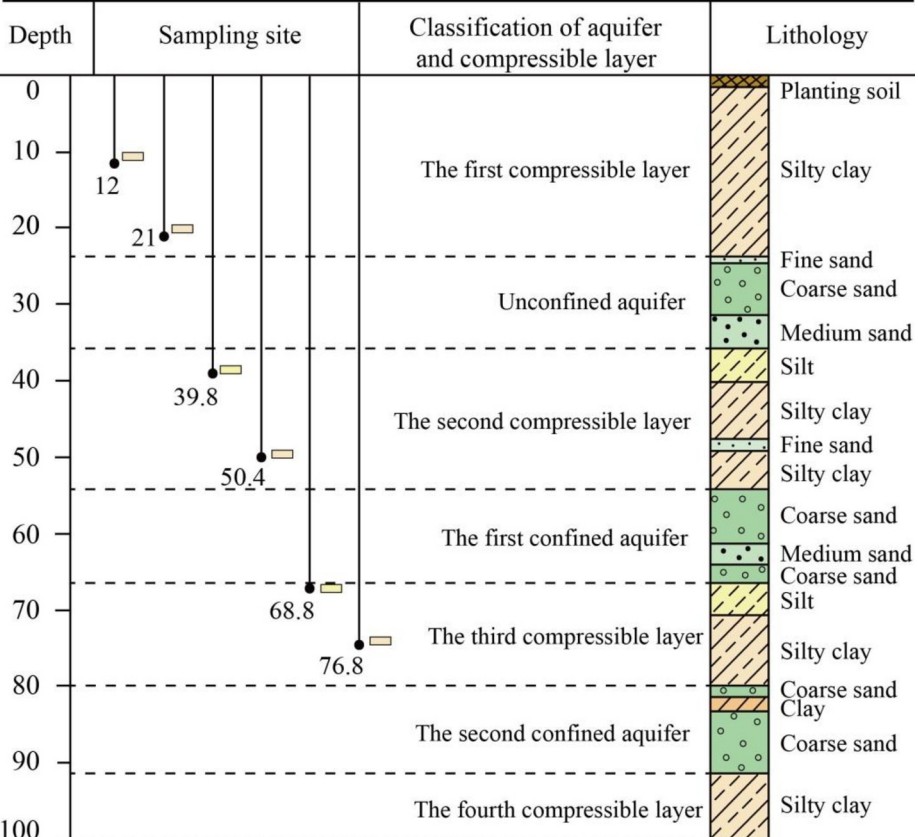

**Fig 2. Schematic description of the borehole BH1, showing the lithological profile in depth of 100 m and vertical location of six soil samples.**

## 3. Experimental design and materials

### 3.1. Experimental materials

The soil samples were all taken from the representative Borehole BH1, located in the subsidence-affected area, as shown in Fig 1. Based on the borehole lithology, the alternating layers within the upper 100 m can be classified into three aquifers that primarily consist of fine sand, medium sand and gravel and four compressible layers that mainly consist of clay and silt, as shown in Fig 2. The compressible layers have a higher compressibility and a greater thickness than the aquifers and are therefore the major contributors to land subsidence [18, 42]. Six soil samples were collected from the three compressible layers at varying depths, and two samples were collected from each layer. The physical and mechanical parameters of all the samples, as determined by relevant ASTM standard test methods [43], are listed in Table 1. All the samples are underconsolidated soils; in particular, the shallow Samples 1 and 2 from the first compressible layer are highly underconsolidated with overconsolidation ratios (*OCRs*), defined as the ratio of preconsolidation stress to current overlying stress, that are much less than 1. Generally, underconsolidated soil is more compressible than normal and overconsolidated soil because it has not yet been fully consolidated [43]. The coefficient of compressibility generally decreases as the depth increases, as listed in Table 1, in which the subscripted number represents the load range. Particularly, the later value indicates the compressibility under actual stress conditions. The variation in the coefficient of compressibility is influenced by the consolidation

**Table 1. Physical and mechanical properties of each soil sample.**

| Sample | Depth (m) | Lithology | Density (g/cm$^3$) | Moisture content (%) | Specific gravity | Void ratio $e_0$ | Coefficient of confined (1D) compressibility $a$ (MPa$^{-1}$) | OCR |
|---|---|---|---|---|---|---|---|---|
| Sample 1 | 12 | Silty clay | 1.93 | 25.8 | 2.72 | 0.77 | $0.30_{0.1-0.2}$ | 0.18 |
| Sample 2 | 21 | Silty clay | 2.12 | 18.1 | 2.71 | 0.51 | $0.21_{0.1-0.2}/0.16_{0.2-0.4}$ | 0.27 |
| Sample 3 | 39.8 | Silt | 2.09 | 17.8 | 2.69 | 0.52 | $0.16_{0.1-0.2}/0.05_{0.4-0.8}$ | 0.73 |
| Sample 4 | 50.8 | Silty clay | 2.02 | 23.1 | 2.72 | 0.66 | $0.20_{0.1-0.2}/0.05_{0.4-0.8}$ | 0.44 |
| Sample 5 | 68.8 | Silt | 2.08 | 18.2 | 2.70 | 0.53 | $0.25_{0.1-0.2}/0.03_{0.8-1.6}$ | 0.95 |
| Sample 6 | 76.8 | Silty clay | 2.00 | 21.3 | 2.71 | 0.64 | $0.29_{0.1-0.2}/0.04_{0.8-1.6}$ | 0.70 |

state, grain size distribution, mineral composition, water content, density, initial porosity and soil structure.

## 3.2. Experimental principle

Land subsidence due to groundwater extraction is a consolidation process that causes a decrease in pore water pressure and an increase in effective stress [43]. In this study, the GDS Consolidation Testing System, developed by Global Digital System Ltd, was used to simulate the change in pore water pressure during the consolidation process in soil samples based on a one-dimensional model [3]. The upper chamber and back-pressure controllers were selected to impose pressures $p_1$ and $p_2$ externally and internally on the samples, as shown in Fig 3. The pressure $p_1$ represents the total stress acting on the samples in the original strata, and the pressure $p_2$ represents the pore water pressure ($u$) inside the sample. Fig 4 presents the change of back pressure and pore water pressure measured by sensors in the experiment, taking Sample 4 for example. It can be verified that the pore water pressure can be absolutely controlled by

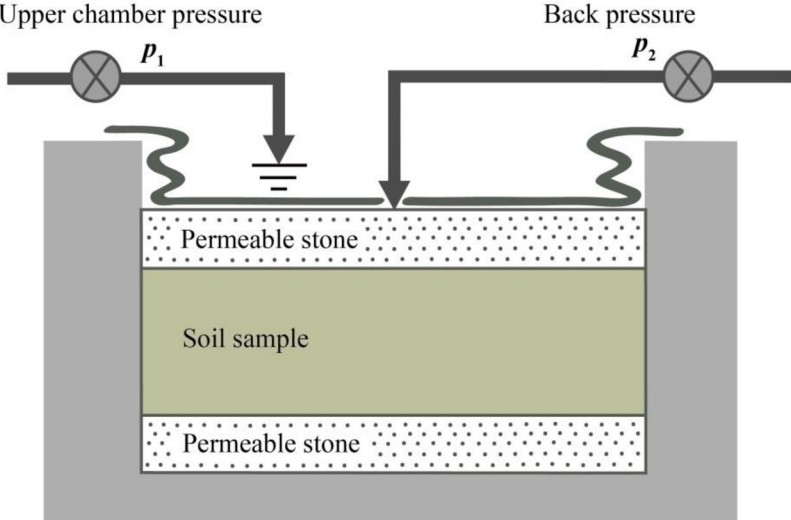

**Fig 3. Schematic illustrating the upper chamber pressure $p_1$ and back pressure $p_2$ act on the soil samples.**

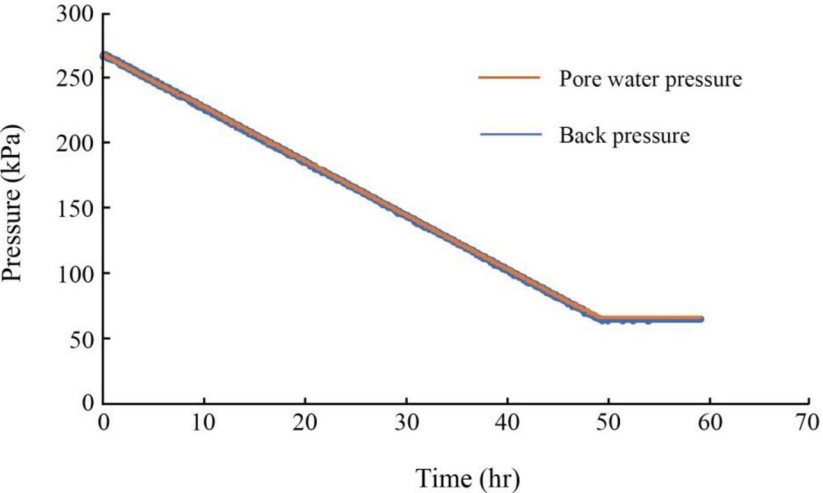

**Fig 4. Change of back pressure and pore water pressure in the withdrawal process, taking sample 4 for example.**

back pressure controller, by controlling the pressures $p_1$ and $p_2$, the simulation of groundwater withdrawal and recharge in the soils under actual stress conditions can be performed.

## 3.3. Experimental design and procedure

Before the laboratory test, all the samples were cut into cylinders of 76.2 mm in diameter and 25 mm in height using a cutting ring as required by the GDS consolidation cell. Then, they were saturated with distilled water using a vacuum method to facilitate the withdrawal and recharge tests.

Taking Sample 5 as an example, the test design is described in Fig 5, and the detailed procedure is listed in Table 2. For each sample, the original overlying stress and pore water pressure

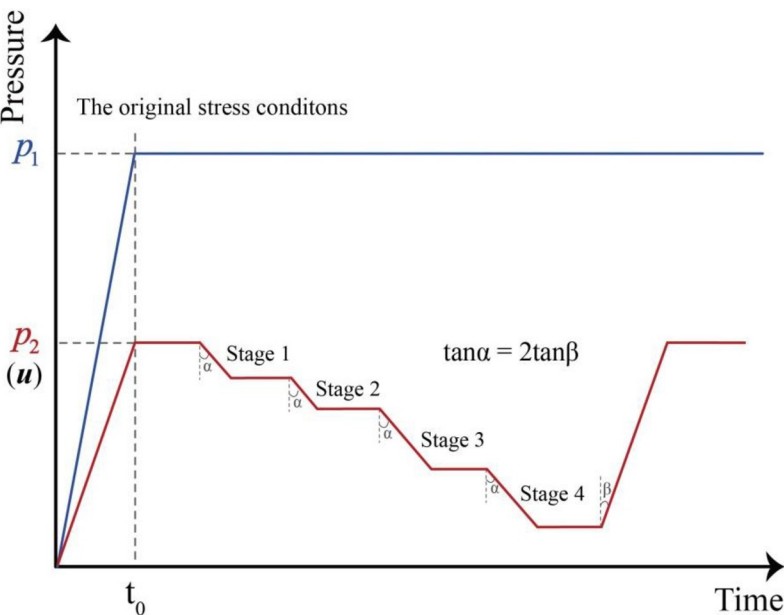

**Fig 5. Schematic diagram of the test design, $P_1$ represents the total stress acting on the samples in the original strata, and $P_2$ represents the pore water pressure ($u$) inside the sample.** Four stages of reducing pore water pressure are designed in the test.

**Table 2. Detailed test procedure for sample 5.**

| Type of test | Sample 5 | | Time duration (min) |
|---|---|---|---|
| | $p_1$ [a] (kPa) | $p_2$ [b] (kPa) | |
| Original stress state | 1456 | 645 | 2880 |
| First reduction stage ($\Delta u_1$) | 1456 | 645–545 | 1440 |
| Stable process | 1456 | 545 | 1440 |
| Second reduction stage ($\Delta u_2$) | 1456 | 545–445 | 1440 |
| Stable process | 1456 | 445 | 1440 |
| Third reduction stage ($\Delta u_3$) | 1456 | 445–245 | 2880 |
| Stable process | 1456 | 245 | 1440 |
| Fourth reduction stage ($\Delta u_4$) | 1456 | 245–45 | 2880 |
| Stable process | 1456 | 45 | 1440 |
| Recovery process | 1456 | 45–645 | 4320 |
| Stable process | 1456 | 645 | >4320 |

[a] $p_1$ represents the total overlying stress

[b] $p_2$ represents the pore water pressure

are first imposed to make the deformation of the soil samples stable under the original stress conditions. Then, the pore water pressures are decreased to simulate the withdrawal process. There is a total of four designed pressure-reduction stages: 0–100 kPa, 100–200 kPa, 200–400 kPa and 400–600 kPa; the water head therefore decreases by 10 m, 20 m, 40 m and 60 m progressively. Notably, shallow soil samples may experience only one or two reduction stages due to their low pore water pressures. The reduction rate is designed to be 100 kPa/d for each stage. When the depressurizing process in each stage is finished, the samples are left for approximately 24 hours to allow any further deformation. After the end of Stage 4, the pore water pressure is gradually increased to the original value at a rate of 200 kPa/d to simulate the recharge process. The detailed test design for each sample is listed in Table 3.

## 4. Results and discussion

### 4.1. Vertical deformation of the soil samples with changing pore water pressure

Fig 6 shows the change in the vertical deformation with changing pore water pressure in the four reduction stages of 0–100 kPa, 100–200 kPa, 200–400 kPa, and 400–600 kPa, named Stage 1, Stage 2, Stage 3 and Stage 4, respectively. At the beginning of each reduction stage, the vertical deformation was reset to zero. The number of soil samples decreases with decreasing pore water pressure (groundwater withdrawal) stage by stage. This is related to the initial pore water pressures of soil samples. In Stage 1, described in Fig 6A, the shallowest sample, i.e.,

**Table 3. Test design for groundwater withdrawal and recharge for each soil sample.**

| Sample | Total stress (kPa) | $\Delta u_1$ (kPa) 0~100 | $\Delta u_2$ (kPa) 100~200 | $\Delta u_3$ (kPa) 200~400 | $\Delta u_4$ (kPa) 400~600 | Recovery process |
|---|---|---|---|---|---|---|
| Sample 1 | 240 | 88–0 | -- | -- | -- | 0–88 |
| Sample 2 | 448 | 176–76 | 76–0 | -- | -- | 0–176 |
| Sample 3 | 844 | 361–261 | 261–161 | 161–0 | -- | 0–361 |
| Sample 4 | 1013 | 464–364 | 364–264 | 264–64 | -- | 64–464 |
| Sample 5 | 1456 | 645–545 | 545–445 | 445–245 | 245–45 | 45–645 |
| Sample 6 | 1570 | 723–623 | 623–523 | 523–323 | 323–123 | 123–723 |

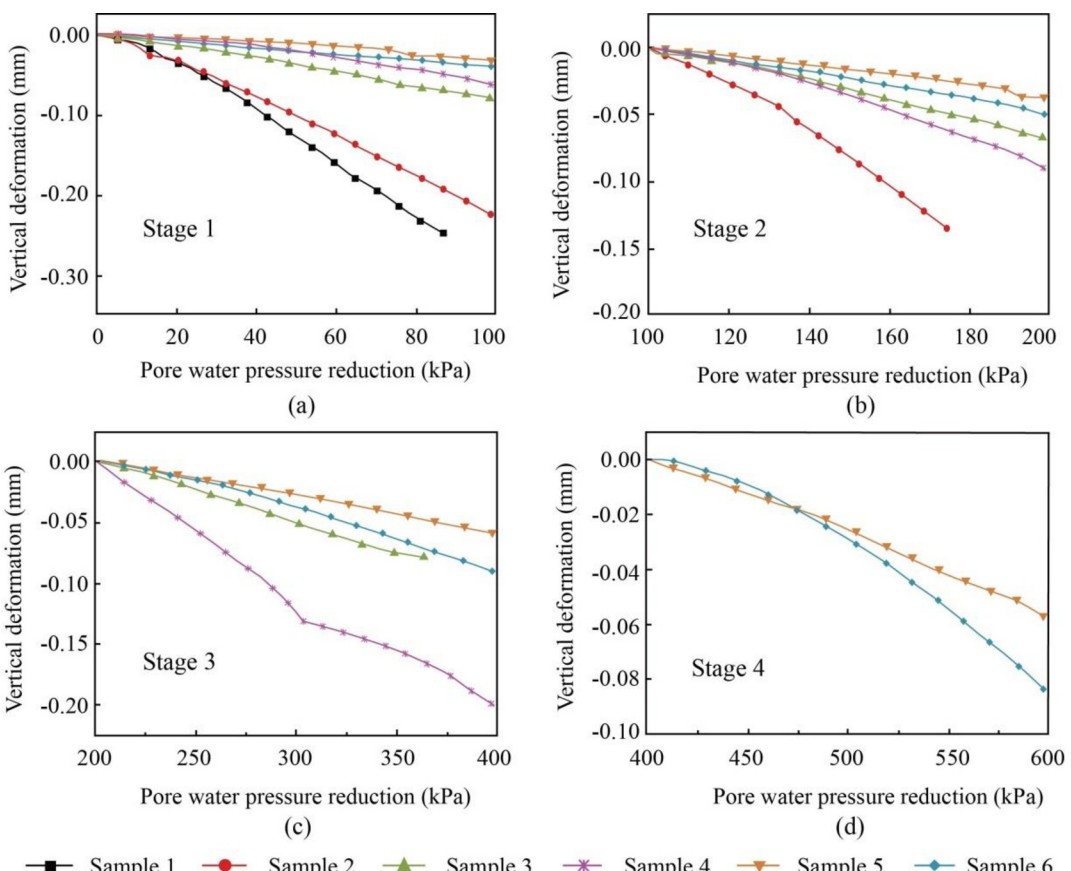

**Fig 6.** Variations in vertical deformation with changing pore water pressure in Stage 1 (a), Stage 2 (b), Stage 3 (c) and Stage 4 (d).

Sample 1 presents the largest deformation when the pressure decreases from 88 to 0 kPa, followed by the thicker Sample 2, with a slightly smaller deformation than Sample 1. The deformations of these samples are both of an order of magnitude greater than those of the other 4 samples. For Sample 1, the unit rate of vertical deformation (deformation rate), defined as the amount of vertical deformation per unit pore water pressure for 1-meter-high samples, is approximately 0.1134 mm/kPa, as shown in Table 4. The deformation rate decreases with

**Table 4. Deformation and corresponding rate in the four reduction stages for each soil sample.**

| Sample | Stage 1 | | Stage 2 | | Stage 3 | | Stage 4 | |
|---|---|---|---|---|---|---|---|---|
| | $S_1$ [a] | $v_1$ [b] | $S_2$ [a] | $v_2$ [b] | $S_3$ [a] | $v_3$ [b] | $S_4$ [a] | $v_4$ [b] |
| Sample 1 | 0.250 | 0.1134 | -- | -- | -- | -- | -- | -- |
| Sample 2 | 0.225 | 0.0900 | 0.135 | 0.0708 | -- | -- | -- | -- |
| Sample 3 | 0.078 | 0.0311 | 0.067 | 0.0269 | 0.078 | 0.0194 | -- | -- |
| Sample 4 | 0.063 | 0.0250 | 0.094 | 0.0378 | 0.201 | 0.0402 | -- | -- |
| Sample 5 | 0.032 | 0.0127 | 0.037 | 0.0148 | 0.057 | 0.0113 | 0.054 | 0.0107 |
| Sample 6 | 0.043 | 0.0172 | 0.047 | 0.0190 | 0.091 | 0.0181 | 0.084 | 0.0168 |

[a] $S$ represents the vertical deformation (mm)

[b] $v$ represents the deformation rate (mm/kPa)

increasing sample depth, which is partially related to the variation in the coefficient of compressibility. However, a high compressibility does not necessarily correspond to a high deformation rate in this study, as exemplified by Samples 4 and 6 in Stage 1. The deformation rate of Sample 6 is smaller than that of Sample 4 despite Sample 6 having a higher compressibility. The deformation variation among the samples in Stages 2, 3 and 4 is similar to that in Stage 1; additionally, the impact of lithology on the soil deformation rate becomes increasingly visible as the pore water pressure decreases stage by stage. Taking Samples 3, composed of silt, and Sample 4, composed of silty clay, as examples, in Stage 1 in Fig 6A, the deformation rate of Sample 4 is smaller than that of Sample 3, whereas in the following two reduction stages in Fig 6B and 6C, the deformation rate of Sample 4 becomes increasingly larger than that of Sample 3. This may be caused by the long duration of clay deformation [26]; the deformation rate of silty clay in the later reduction stage is more prone to be influenced by the deformation in the preceding reduction stage.

For the same sample in the four reduction stages, it can be observed that as the pore water pressure decreases stage by stage, the deformation rate decreases gradually, as summarized in Table 4. However, for the deeper samples composed of silty clay, such as Samples 4 and 6, the deformation rate remains unchanged or even increases from Stage 1 to Stage 4, which can also be explained by the long duration of clay deformation.

Land subsidence is a coupled process of groundwater flow and soil deformation. The theoretical model of land subsidence in this paper is based on Terzaghi's 1D consolidation theory, the compression of soil layer $S$ can be expressed by the following equation [44], which can be used for validation for the experimental data.

$$S = m_V \Delta\sigma' H = \frac{a\Delta\sigma'}{1 + e_0} H \tag{1}$$

where $m_v$ is the volumetric compressibility, $\Delta\sigma'$ is the increase of effective stress, which is the decrease of pore water pressure $\Delta u$, $H$ is the thickness of the soil layer. The parameter $m_v$ can be derived by the coefficient of confined (1D) compressibility $a$ and void ratio $e_0$.

The theoretical result for each soil sample is illustrated in Table 5. The parameters $a$ are the values under actual stress conditions listed in Table 1. The theoretical calculation merely aimed at Stage 1 because the deformation in Stage 1 may not been affected by the deformation in the previous stages. The theoretical deformation is basically consistent with the experimental result, which verifies the reliability of the experiment.

## 4.2. Hysteresis effect caused by water withdrawal

The hysteresis effect caused by water withdrawal is a complex issue that can involve several factors, such as leaking recharge from aquitards to aquifers, internal overpressure disequilibria in soils, and creep effects [45–47]. However, in the experiment, the hysteresis only involves the creep behavior of soil, which is one form of secondary consolidation. The creep deformations

**Table 5. Comparison of experimental and theoretical deformation.**

| Samples | Experimental deformation (mm) | Theoretical deformation (mm) | $\Delta u_1$ (kPa) |
|---|---|---|---|
| Sample 1 | 0.250 | 0.298 | 88–0 |
| Sample 2 | 0.225 | 0.220 | 100–0 |
| Sample 3 | 0.078 | 0.072 | 100–0 |
| Sample 4 | 0.063 | 0.061 | 100–0 |
| Sample 5 | 0.032 | 0.039 | 100–0 |
| Sample 6 | 0.043 | 0.048 | 100–0 |

**Table 6. Creep deformation and ratio in the four reduction stages for each soil sample.**

| Sample | Stage 1 | | Stage 2 | | Stage 3 | | Stage 4 | |
|---|---|---|---|---|---|---|---|---|
| | $H_1$ [a] | $HR_1$ [b] | $H_2$ [a] | $HR_2$ [b] | $H_3$ [a] | $HR_3$ [b] | $H_4$ [a] | $HR_4$ [b] |
| Sample 1 | 0.040 | 16.00 | -- | -- | -- | -- | -- | -- |
| Sample 2 | 0.040 | 17.78 | 0.060 | 44.38 | -- | -- | -- | -- |
| Sample 3 | 0.012 | 15.42 | 0.019 | 28.27 | 0.022 | 28.20 | -- | -- |
| Sample 4 | 0.025 | 39.94 | 0.040 | 42.37 | 0.045 | 22.39 | -- | -- |
| Sample 5 | 0.002 | 6.31 | 0.004 | 10.81 | 0.014 | 24.69 | 0.015 | 27.98 |
| Sample 6 | 0.007 | 17.44 | 0.010 | 21.01 | 0.019 | 20.97 | 0.025 | 29.79 |

[a] $H$ represents the creep deformation (mm)

[b] $HR$ represents the creep ratio (%)

of soil samples within 24 hours after each reduction stage and the corresponding creep ratios between the secondary compression and deformation in the previous reduction stage are summarized in Table 6. The variations in secondary compression (creep) among different samples correspond well with the changes in deformation in the previous reduction stage of pore water pressure. In the same reduction stage, the hysteresis effect of shallow soils is more obvious than that of deep soils with the same lithology. In particular, the creep deformation of Samples 1 and 2 is 0.04 mm, one to two orders of magnitude greater than that of Samples 4 and 6. In the same compressible layer, silty clays, such as in Samples 4 and 6, exhibit larger creep deformation and corresponding ratios than silts, such as in Samples 3 and 5, in different reduction stages. For the same soil sample, with the pore water pressure decreasing stage by stage, the creep deformation and ratio show an overall increasing trend, indicating that the hysteresis effect becomes more apparent and that the secondary compression plays a more important role with increasing groundwater withdrawal. This pattern is shown in Fig 7, taking Sample 4 as an example.

## 4.3. Rebound deformation caused by water recharge

When all the samples were left for 24 hours after the last reduction stage, the pore water pressures began to increase to the original value at a rate of 200 kPa/d to simulate the groundwater

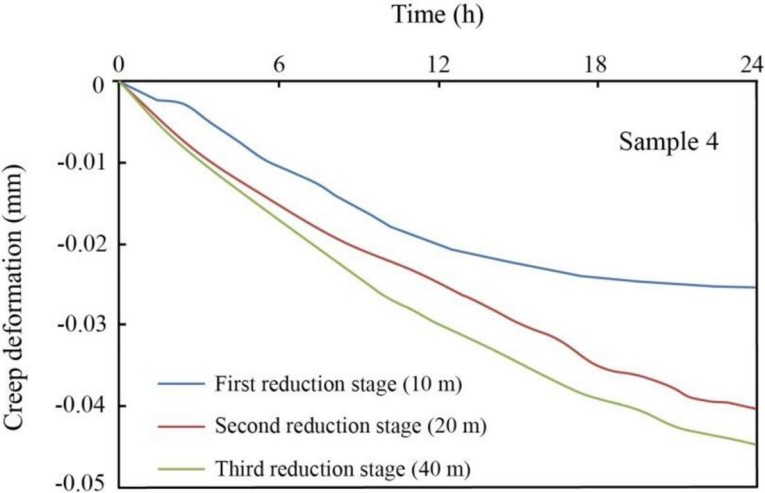

**Fig 7. Creep deformation within 24 hours after each reduction stage, taking sample 4 as an example.**

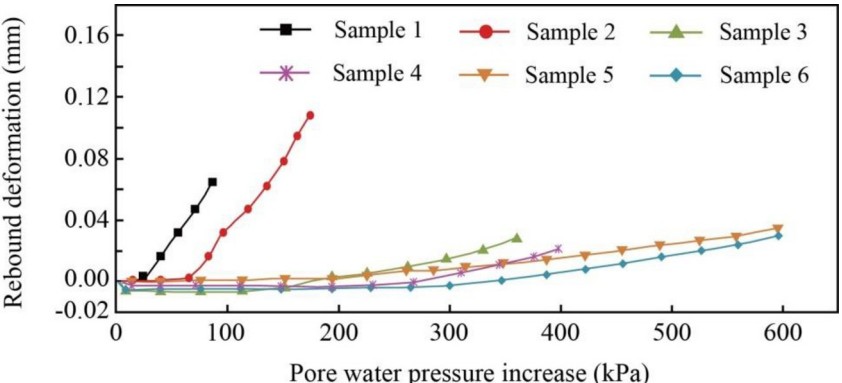

**Fig 8. Rebound deformation with increasing pore water pressure for each soil sample.**

recharge. Fig 8 presents the rebound curve for each soil sample, and the rebound deformations and corresponding ratios of rebound to total deformation are summarized in Table 7. The deformation was set to zero when the pore water pressure began to increase. It can be observed that all the soil samples are characterized by elastic-plastic deformation. There is a lag between the rebound and the increase in the pore water pressure, particularly for some individual samples, the compression even continued in the beginning. The lag varies with the decrease in pore water pressure. For the deep Samples 5 and 6, the pore water pressure decreased by 600 kPa over the four reduction stages, and the soils did not rebound until the pore water pressure increased to approximately 300 kPa. For the shallow Samples 1 and 2, the pore water pressure decreased by 88.2 kPa and 176.4 kPa, and the rebound curves showed considerable increases at approximately 25 kPa and 60 kPa. The lag may be caused by the unstable deformation of soils prior to the increase in pore water pressure, so compaction will continue for a time during the rebound process. The rebound deformations of shallow Samples 1 and 2 are 0.07 mm and 0.112 mm, respectively, in the process of increasing the pore water pressure, accounting for 24.1% and 24.3% of the total deformation; these values are greater than those of Samples 4 and 6 with the same lithology. In the same compressible layer, the rebound deformation and ratio of silty clay, such as in Samples 4 and 6, is smaller than that of silt, such as in Samples 3 and 5.

In the experiment, the increasing pore water pressure is an unloading process. The preconsolidation stress is an important factor influencing the rebound deformation of soils. For effective stress changes that are less than the preconsolidation stress, the compaction or rebound of the soils is elastic and is fully recoverable under stress; for effective stresses that are larger than the preconsolidation stress, the virgin compaction of the soils is chiefly inelastic, with a small recoverable elastic component [26]. The soil samples in the experiment are all underconsolidated; therefore, they experience virgin compaction in the process of decreasing pore water

Table 7. Rebound and corresponding ratio for each soil sample.

| Sample | Rebound (mm) | Rebound ratio (%) |
|---|---|---|
| Sample 1 | 0.070 | 24.14 |
| Sample 2 | 0.112 | 24.34 |
| Sample 3 | 0.030 | 10.87 |
| Sample 4 | 0.020 | 4.27 |
| Sample 5 | 0.035 | 16.41 |
| Sample 6 | 0.030 | 9.46 |

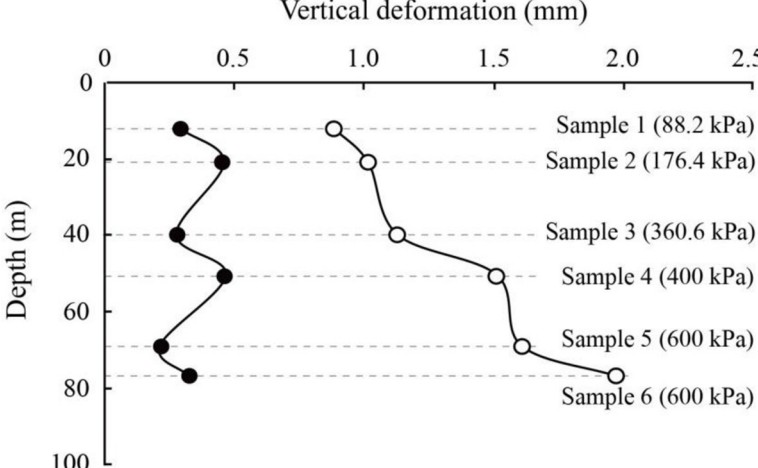

**Fig 9. Comparison in vertical deformations under two types of loading, the solid circles represent the deformation in the test, and hollow circles represent the deformation in routine compression test.**

pressure, with a small amount of rebound deformation. The rebound variation among the soil samples is also influenced by the rebound index of the soil. He et al. [48] noted that there is a linear relationship between the compression index $C_c$ and the rebound index $C_s$ of Shanghai clay. The ratio between $C_c$ and $C_s$ varies in the range of 4.8 to 6.9 for normally consolidated soils and 3.3 to 5.2 for overconsolidated soils. Therefore, a more compressible soil, such as that in Samples 1 and 2, is more likely to rebound during unloading than Samples 4 and 6. For the soils with different lithologies, the deformation of clay is less recoverable than that of silt due to the distinct granular components and pore distributions. Overall, the rebound of the soil samples in the experiment is a complex issue that involves several factors, such as preconsolidation stress, the rebound index, lithology, and previous reductions in pore water pressure.

## 4.4. Discussion

The experiment conducted in this paper is verified to be reliable. It has been successfully applied in the newly developed area of Tongchuan region, located north of Xi'an city, Shaanxi Province, where land subsidence becomes a serious geological problem since 2000 due to the over exploitation of groundwater [3]. The experiment can also be applied to other regions to simulate the groundwater withdrawal and recharge. However, the lithology of soils should be clay, silty clay, silt, fine or medium sand. The soils mixed with coarse sand and clays such as debris soils may be limited in the experiment because the large difference in particle properties will influence the accuracy of the experimental result.

In the experiment, the groundwater withdrawal and recharge are simulated by decreasing and increasing the pore water pressure. The decrease in pore water pressure, accompanied by an increase in effective stress, induces the consolidation of the soils. However, a routine compression test cannot reflect the soil deformation caused by changes in pore water pressure. This is because the pressures in a routine compression tests are all given in a moment [43], whereas groundwater withdrawal is a gradual process. The pore water pressure in the experiment in this paper varies over a period with a designed rate. The two types of loading induce quite different deformations, as presented in Fig 9. For the same soil sample, the deformation in the routine compression test is much larger than that in the experiment with the same change in pressure. In addition, the deformation variations among soil samples are also

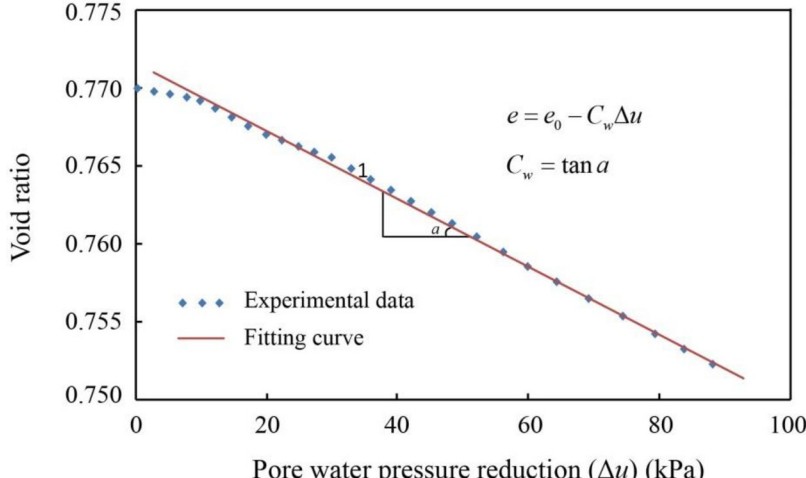

**Fig 10. Linear subsidence model showing the variation in void ratio with pore water pressure reduction.**

different in the two tests. For example, the total deformation of Sample 3 is slightly smaller than that of Sample 1 in the experiment in this paper, whereas it is much larger than that of Sample 1 in the routine compression test.

Therefore, the coefficient of compressibility determined by the routine compression test cannot be taken as the optimal parameter for describing the soil deformation caused by the variation in pore water pressure. A parameter $C_w$, named the subsidence index, is proposed in this paper [4]. In Fig 6A, when the pore water pressure decreases by 100 kPa, corresponding to a decrease in the groundwater of 10 m, the change in the vertical deformation with changing pore water pressure shows an overall linear distribution. Therefore, a linear subsidence model is proposed, as shown in Eq 2 and in Fig 10, taking Sample 3 as an example.

$$e = e_0 - C_w \Delta u \tag{2}$$

where $e_0$ is the initial void ratio; $\Delta u$ is the decrease in pore water pressure, ranging from 0 to 100 kPa; and $C_w$ is the subsidence index, reflecting the soil compressibility during withdrawal. This model is not only influenced by consolidation states, grain size composition, mineral composition, water content, density, initial porosity and soil structure, as is the coefficient of compressibility, but also varies with the withdrawal pattern, such as the withdrawal rate. The larger the parameter $C_w$ is, the more likely the soils are to experience compression. The $C_w$ values for the six samples are listed in Table 8.

The alternating layers at depths of less than 100 m in the subsidence-affected area of Changping District can be classified into three aquifers and four compressible layers. The

**Table 8. Subsidence index $C_w$ of each soil sample.**

| Sample | Initial void ratio $e_0$ | Void ratio $e$ | Subsidence index $C_w$ (kPa$^{-1}$) |
|---|---|---|---|
| Sample 1 | 0.77 | 0.7523 | 2.01E-4 |
| Sample 2 | 0.51 | 0.4964 | 1.36E-4 |
| Sample 3 | 0.52 | 0.5152 | 4.74E-5 |
| Sample 4 | 0.66 | 0.6558 | 4.18E-5 |
| Sample 5 | 0.53 | 0.5280 | 1.96E-5 |
| Sample 6 | 0.64 | 0.6372 | 2.82E-5 |

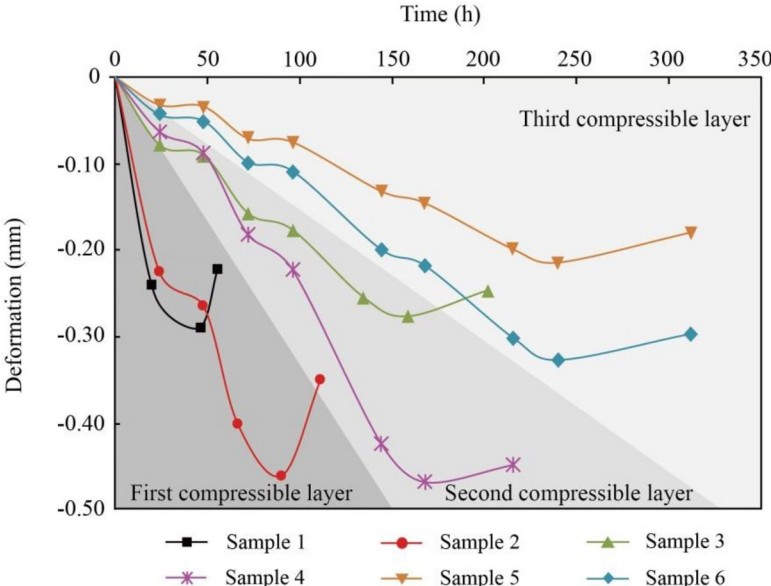

**Fig 11. Deformation in the whole process of withdrawal and recharge for each soil sample.**

compressible layers are the main contributor to land subsidence due to their high compressibility and great thickness. Compression in compressible layers is usually caused by recharge to the adjacent aquifers; therefore, the decrease in the pore water pressure and the corresponding compression in compressible layers is a gradual and long-term process. Six soil samples were all taken from the compressible layers. The first aquifer below the first compressible layer is usually taken as the optimal water resource for daily life and industrial and agricultural production because of the shallow depth and high water yield. The first compressible layer above is therefore easily influenced by groundwater exploitation due to recharge. However, Samples 1 and 2 from the first compressible layer show large deformation and considerable hysteresis in the same pore water pressure reduction stage compared with other soil samples from the second and third compressible layers, as illustrated in Fig 11. Therefore, the vertical deformation in the first compressible layer under withdrawal should be paid great attention to. In the same compressible layer, the silty clay is more compressible and hysteretic than the silt, and deformation of the silty clay is less recoverable than that of the silt. The second and third compressible layers composed of these two types of soils show less deformation than the first compressible layer, particularly in Stage 1 and Stage 2. However, when groundwater in the adjacent aquifer is exploited on a large scale, that is, the pore water pressure decreases dramatically, the hysteresis effect is particularly apparent, as shown in Fig 11. The deformation is therefore a gradual and long-term process. In addition, there is a long lag for the soils to begin to rebound when the groundwater is recharged, and the rebound is extremely small compared with the large deformation during exploitation, as presented by Samples 4 to 6 in Fig 11. Therefore, the compression in the second and third compressible layers caused by water exploitation in the adjacent aquifers also cannot be ignored.

The soil deformation caused by groundwater withdrawal and recharge is influenced not only by the properties of the soils but also by the withdrawal/recharge patterns. Therefore, changing the patterns in the subsequent study is necessary to help us select the best solution to mitigate the land subsidence.

## 5. Conclusions

In this paper, experimental tests were performed on six soil samples with various lithologies taken from different depths in Changping District of Beijing to characterize the soil deformations caused by groundwater withdrawal and recharge.

In the same pore water pressure reduction stage, shallow soils show greater deformation and more apparent hysteresis effect than deeper soils with the same lithology. In the same compressible layer, silty clay is more compressible and hysteretic than silt. For the same soil sample, the deformation rate decreased gradually as the pore water pressure decreased stage by stage, whereas the creep deformation shows an overall increasing trend. All the soil samples are characterized by elastic-plastic deformation, and the shallow soil samples with less pore water pressure decrease are more likely to exhibit rebound.

A parameter named the subsidence index $C_w$ is proposed in this paper to describe the soil compressibility under withdrawal. It is influenced by both the soil properties and the withdrawal pattern. The larger the parameter $C_w$ is, the more likely the soils are to experience compression.

The alternating layers at depths of less than 100 m in the subsidence-affected area of Changping District can be classified into three aquifers and four compressible layers. The compressible layers are the main contributors to land subsidence due to their high compressibility and great thickness. During a given pore water pressure reduction stage, the first compressible layer is characterized by greater deformation and more considerable hysteresis than the other compressible layers. Therefore, the exploitation in the adjacent aquifer should be controlled in the future. When the groundwater in the aquifers below the second and third compressible layers is exploited on a large scale, the deformation in the two compressible layers presents a gradual and long-term process with little rebound, which should also be paid great attention to.

The experiment conducted in this paper is verified to be reliable. It can be applied to other regions to simulate the groundwater withdrawal and recharge. However, the lithology of soils should be clay, silty clay, silt, fine or medium sand. The soils mixed with coarse sand and clays such as debris soils may be limited.

## Author Contributions

**Conceptualization:** Yanbo Cao, Ya-ni Wei.

**Data curation:** Yanbo Cao.

**Formal analysis:** Ya-ni Wei.

**Funding acquisition:** Yanbo Cao, Min Peng, Liangliang Bao.

**Investigation:** Yanbo Cao.

**Validation:** Wen Fan.

**Writing – original draft:** Ya-ni Wei, Wen Fan.

**Writing – review & editing:** Yanbo Cao, Ya-ni Wei.

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
