## [Decision Letter · Decision Letter 0]

2 Mar 2020

PONE-D-20-02953

Experimental study of land subsidence in response to groundwater withdrawal and recharge in Changping District of Beijing

PLOS ONE

Dear Mrs Wei,

Thank you for submitting your manuscript to PLOS ONE. After careful consideration, we feel that it has merit but does not fully meet PLOS ONE’s publication criteria as it currently stands. Therefore, we invite you to submit a revised version of the manuscript that addresses the points raised during the review process.

We would appreciate receiving your revised manuscript by Apr 16 2020 11:59PM. To enhance the reproducibility of your results, we recommend that if applicable you deposit your laboratory protocols in protocols.io, where a protocol can be assigned its own identifier (DOI) such that it can be cited independently in the future. For instructions see: http://journals.plos.org/plosone/s/submission-guidelines#loc-laboratory-protocols

We look forward to receiving your revised manuscript.

Kind regards,

Jianguo Wang, PhD

Academic Editor

PLOS ONE

Journal Requirements:

2. We note that your model uses the Changping District, but you do not discuss whether the method can be successfully applied to other regions. Please amend your Discussion section and Conclusions to provide a detailed discussion of whether the method is generalizable and can be applied to other regions (and if so, are there restrictions on what type of region?) and include any foreseeable limitations.

4. We note that Figure #1 in your submission contain map images which may be copyrighted. All PLOS content is published under the Creative Commons Attribution License (CC BY 4.0), which means that the manuscript, images, and Supporting Information files will be freely available online, and any third party is permitted to access, download, copy, distribute, and use these materials in any way, even commercially, with proper attribution. For these reasons, we cannot publish previously copyrighted maps or satellite images created using proprietary data, such as Google software (Google Maps, Street View, and Earth). For more information, see our copyright guidelines: http://journals.plos.org/plosone/s/licenses-and-copyright.

1.    You may seek permission from the original copyright holder of Figure #1 to publish the content specifically under the CC BY 4.0 license. 

Reviewers' comments:

Reviewer's Responses to Questions

**Comments to the Author**

1. Is the manuscript technically sound, and do the data support the conclusions?

Reviewer #1: Partly

Reviewer #2: Yes

2. Has the statistical analysis been performed appropriately and rigorously? 

Reviewer #1: No

Reviewer #2: Yes

3. Have the authors made all data underlying the findings in their manuscript fully available?

Reviewer #1: No

Reviewer #2: Yes

4. Is the manuscript presented in an intelligible fashion and written in standard English?

Reviewer #1: Yes

Reviewer #2: Yes

5. Review Comments to the Author

Reviewer #1: The authors did experiments on compressibility layers due to land subsidence in the Changping district of Beijing city, China. I hold two major concerns about this manuscript.

1- Can the authors show the trend of the groundwater depletion in this district, though recent in situ measurements?

2- Can the authors show validation for their results using independent techniques (eg., InSAR, levelling, GPS) for this area

Minors

In the introduction section, no mention in using the GPS data to characterize the land subsidence; there are plenty of studies that adopted this technique along with InSAR and levelling methods.

Reviewer #2: Assessment of paper on

Experimental study of land subsidence in response to groundwater withdrawal and

recharge in Changping District of Beijing

This study presents a series of experimental tests using the GDS Consolidation Testing System to characterize the compression and rebound of soils caused by groundwater withdrawal and recharge and proposes a parameter named the subsidence index Cw to describe the soil compressibility during groundwater withdrawal. The paper is very well written. The literature review in the introduction is thorough and the objectives are clearly stated. The methodology is explained in detail and the results are nicely reported and discussed. The manuscript could be accepted after intensive revision with clarification of the following comments

1) Lines 121 to 123, the study mainly focus on the the compressible layers, which is reasonable to certain degrees. However, the pumped aquifer or the aquitard, which contributes more for the land subsidence is debatable in different areas. I think a reference proving the relationship between the four compressible layers and land subsidence is necessary.

2) Lines 148 to 152, the study simulates groundwater withdrawal and recharge in the soils by controlling the pressures p1 and p2. I think a figure illustrating how p1 and p2 act on soil sample can help readers better understand the simulation of groundwater withdrawal and recharge.

3) It seems that the pore pressure in soil sample changes immediately as the back-pressure changes according to the authors’ description. This may be reasonable for sandy soil or silty clay with higher permeability. Have you ever monitored the pore pressure change in the sample center?

4) Please explain how to obtain the creep deformation of soil sample. If there is a hysteresis between the pore pressure at sample center and back-pressure, how to distinguish the consolidation deformation and creep deformation?

5) Lines 192 to 198, please explain why a high compressibility does not necessarily correspond to a high deformation rate?

6. PLOS authors have the option to publish the peer review history of their article (what does this mean?). If published, this will include your full peer review and any attached files.

Reviewer #1: No

Reviewer #2: No

---

## [Author Response · Author response to Decision Letter 0]

14 Apr 2020

We would like to thank reviewers for their insightful comments on the manuscript, as these comments led us to a great improvement of the work. Our revisions reflect all reviewers’ suggestions and comments. Detailed responses to reviewers are given below. 

Journal Requirements 

RESPONSE: Yes, we have revised the manuscript style as required by PLOS ONE. 

2) We note that your model uses the Changping District, but you do not discuss whether the method can be successfully applied to other regions. Please amend your Discussion section and Conclusions to provide a detailed discussion of whether the method is generalizable and can be applied to other regions (and if so, are there restrictions on what type of region?) and include any foreseeable limitations.

RESPONSE: The experiment conducted in this paper has been successfully applied in the newly developed area of Tongchuan region, located north of Xi’an city, Shaanxi Province, where land subsidence becomes a serious geological problem since 2000 due to the over exploitation of groundwater [1]. The experiment can also be applied to other regions to simulate the groundwater withdrawal and recharge. However, the lithology of soils should be clay, silty clay, silt, fine or medium sand. The soils mixed with coarse sand and clays such as debris soils may be limited in the experiment because the large difference in particle properties will influence the accuracy of the experimental result. The application and limitation of the experiment conducted in this paper has been added in the Discussion and Conclusion, seen in line 331-339, 440-443 in “Revised Manuscript with Track Changes”

[1] Wei, Y. N.; Fan, W.; Cao, Y. B. Experimental study on the vertical deformation of aquifer soils under conditions of withdrawing and recharging of groundwater in Tongchuan region, China. Hydrogeol. J. 2017, 25, 297-309.

3) In your Methods section, please provide additional information regarding the permits you obtained for the work. Please ensure you have included the full name of the authority that approved the field site access and, if no permits were required, a brief statement explaining why.

RESPONSE: The sampling area was neither cultivated nor urban land, it was a wasteland. Therefore, no permits were required for the work.

4) We note that Figure #1 in your submission contain map images which may be copyrighted. All PLOS content is published under the Creative Commons Attribution License (CC BY 4.0), which means that the manuscript, images, and Supporting Information files will be freely available online, and any third party is permitted to access, download, copy, distribute, and use these materials in any way, even commercially, with proper attribution. For these reasons, we cannot publish previously copyrighted maps or satellite images created using proprietary data, such as Google software (Google Maps, Street View, and Earth). For more information, see our copyright guidelines: http://journals.plos.org/plosone/s/licenses-and-copyright.

RESPONSE: The topographic map of Beijing and hydrological profile in Figure 1 are referred to others and we have deleted them in Figure 1. The geological map of the study area with Quaternary thickness contour line was drawn by Yanbo Cao referring to available data. Therefore, the copyright is ours.

Reviewer #1

The authors did experiments on compressibility layers due to land subsidence in the Changping district of Beijing city, China. I hold two major concerns about this manuscript.

1) Can the authors show the trend of the groundwater depletion in this district, though recent in situ measurements?

RESPONSE: Referring to the monitoring data of Water Bureau of Changping District [2], the shallow groundwater depth in the monitoring point of Machikou (marked in Figure 1) presented an increasing trend from 6.09 m in 2001 to 28.09 m in 2011, as shown below. During the 10 years, the decline of groundwater level is more than 20 m, and the decline rate is an average of 2 m per year. The trend of groundwater depletion has been added in the section “Geological and hydrogeological setting of the study area”, seen in line 111-115 in “Revised Manuscript with Track Changes” 

Figure. 1 The change of groundwater depth from 2001 to 2011 (Revised after Liu et al.)

[2] Liu, E. M.; Zhang, N.; Han, P. L. Analysis of present situation of groundwater level in Changping District based on GIS. Beijing Water 2014, 4, 24-27. (in Chinese)

2) Can the authors show validation for their results using independent techniques (eg., InSAR, levelling, GPS) for this area

RESPONSE: In the experiment, the variation of hydraulic head is controlled by the back pressure controller through pore water pressure, and the variation design is primarily used for prediction of land subsidence in the future, which is not consistent with the actual variation of hydraulic head in the area, therefore, it is hard to make a contrast of land subsidence between the experimental results and field measurement using techniques such as InSAR, leveling and GPS. However, the theoretical calculation has been conducted in the last part of Section 4.1 based on Terzaghi’s 1D consolidation theory to verify the experimental results, seen in line 240-255 in “Revised Manuscript with Track Changes”. Results showed that the theoretical deformation are basically consistent with the experimental results, which verify the reliability of the experiment in this paper. 

3) In the introduction section, no mention in using the GPS data to characterize the land subsidence; there are plenty of studies that adopted this technique along with InSAR and levelling methods.

RESPONSE: Yes, GPS is also an important technique to characterize the land subsidence. The technique and two corresponding references have been added in the Introduction section, seen in line 48 in “Revised Manuscript with Track Changes”

Reviewer #2

1) This study presents a series of experimental tests using the GDS Consolidation Testing System to characterize the compression and rebound of soils caused by groundwater withdrawal and recharge and proposes a parameter named the subsidence index Cw to describe the soil compressibility during groundwater withdrawal. The paper is very well written. The literature review in the introduction is thorough and the objectives are clearly stated. The methodology is explained in detail and the results are nicely reported and discussed. The manuscript could be accepted after intensive revision with clarification of the following comments

1) Lines 121 to 123, the study mainly focus on the compressible layers, which is reasonable to certain degrees. However, the pumped aquifer or the aquitard, which contributes more for the land subsidence is debatable in different areas. I think a reference proving the relationship between the four compressible layers and land subsidence is necessary.

RESPONSE: Two references have been added，as seen in line 129 in “Revised Manuscript with Track Changes”. Referring to the references, the layers composed of clays and silty clays are the main compressible layers for land subsidence in Changping District as well as Beijing city.

2) Lines 148 to 152, the study simulates groundwater withdrawal and recharge in the soils by controlling the pressures p1 and p2. I think a figure illustrating how p1 and p2 act on soil sample can help readers better understand the simulation of groundwater withdrawal and recharge.

RESPONSE: Yes, a figure named Figure 3 in the text that illustrates how p1 and p2 act on soil sample has been provided. 

3) It seems that the pore pressure in soil sample changes immediately as the back-pressure changes according to the authors’ description. This may be reasonable for sandy soil or silty clay with higher permeability. Have you ever monitored the pore pressure change in the sample center?

RESPONSE: Yes, the pore water pressure is also monitored by sensors in GDS Consolidation Testing System. A figure named Figure 4 that presents the change of back pressure and pore water pressure of Sample 4 with lithology of silty clay is provided in the test. It can be seen that pore water pressure and back pressure match well, indicating that the pore water pressure can be absolutely controlled by back pressure controller, seen in line 157-160 in “Revised Manuscript with Track Changes”. 

4) Please explain how to obtain the creep deformation of soil sample. If there is a hysteresis between the pore pressure at sample center and back-pressure, how to distinguish the consolidation deformation and creep deformation?

RESPONSE: The creep deformation in this paper is the deformation of soil samples after each reduction stage of pore water pressure, that is the deformation when the pore water pressure has been decreased and kept constant. we monitored the creep deformation within 24 hours when the pore water keep constant to observe the hysteresis effect of soil samples. There is no hysteresis between pore water pressure and back pressure referring to the monitoring data as presented in Figure 4 in the text. Therefore, the consolidation deformation and creep deformation can be distinguished clearly. 

5) Lines 192 to 198, please explain why a high compressibility does not necessarily correspond to a high deformation rate?

RESPONSE: Soil compressibility is influenced by various factors such as the consolidation state, grain size distribution, mineral composition, water content, density, initial porosity and soil structure. The permeability can largely determine the soil deformation. However, the coefficient of permeability is usually obtained by compression test, in which the pressures are all given in moment. As for the experiment in this paper, the increase in effective stress and decrease in pore water pressure is a gradual process. Therefore, the deformation of soil samples in the experiment is influenced not only by permeability but also by the loading process. Therefore, in the section of Discussion, the parameter subsidence index Cw is proposed, which is not only influenced by consolidation states, grain size composition, mineral composition, water content, density, initial porosity and soil structure, as is the coefficient of compressibility, but also varies with the withdrawal pattern, such as the withdrawal rate.

---

## [Decision Letter · Decision Letter 1]

23 Apr 2020

Experimental study of land subsidence in response to groundwater withdrawal and recharge in Changping District of Beijing

PONE-D-20-02953R1

Dear Dr. Wei,

We are pleased to inform you that your manuscript has been judged scientifically suitable for publication and will be formally accepted for publication once it complies with all outstanding technical requirements.

With kind regards,

Jianguo Wang, PhD

Academic Editor

PLOS ONE

Additional Editor Comments (optional):

Reviewers' comments:

Reviewer's Responses to Questions

**Comments to the Author**

1. If the authors have adequately addressed your comments raised in a previous round of review and you feel that this manuscript is now acceptable for publication, you may indicate that here to bypass the “Comments to the Author” section, enter your conflict of interest statement in the “Confidential to Editor” section, and submit your "Accept" recommendation.

Reviewer #1: All comments have been addressed

Reviewer #2: All comments have been addressed

2. Is the manuscript technically sound, and do the data support the conclusions?

Reviewer #1: Yes

Reviewer #2: (No Response)

3. Has the statistical analysis been performed appropriately and rigorously? 

Reviewer #1: Yes

Reviewer #2: (No Response)

4. Have the authors made all data underlying the findings in their manuscript fully available?

Reviewer #1: Yes

Reviewer #2: Yes

5. Is the manuscript presented in an intelligible fashion and written in standard English?

Reviewer #1: Yes

Reviewer #2: Yes

6. Review Comments to the Author

Reviewer #1: (No Response)

Reviewer #2: (No Response)

7. PLOS authors have the option to publish the peer review history of their article (what does this mean?). If published, this will include your full peer review and any attached files.

Reviewer #1: No

Reviewer #2: No

---

## [Editor Report · Acceptance letter]

30 Apr 2020

PONE-D-20-02953R1 

Experimental study of land subsidence in response to groundwater withdrawal and recharge in Changping District of Beijing 

Dear Dr. Wei:

I am pleased to inform you that your manuscript has been deemed suitable for publication in PLOS ONE. Congratulations! Your manuscript is now with our production department. 

With kind regards,

on behalf of

Dr. Jianguo Wang 

Academic Editor

PLOS ONE